# A new interhemispheric teleconnection increases predictability of winter precipitation in southwestern US

Antonios Mamalakis [1], Jin-Yi Yu [2], James T. Randerson [2], Amir AghaKouchak [1,2] & Efi Foufoula-Georgiou [1,2]

Reliable prediction of seasonal precipitation in the southwestern US (SWUS) remains a challenge with significant implications for the economy, water security and ecosystem management of the region. Winter precipitation in the SWUS has been linked to several climate modes, including the El Niño-Southern Oscillation (ENSO), with limited predictive ability. Here we report evidence that late-summer sea surface temperature and geopotential height anomalies close to New Zealand exhibit higher correlation with SWUS winter precipitation than ENSO, enhancing the potential for earlier and more accurate prediction. The teleconnection depends on a western Pacific ocean-atmosphere pathway, whereby sea surface temperature anomalies propagate from the southern to the northern hemisphere during boreal summer. Analysis also shows an amplification of this new teleconnection over the past four decades. Our work highlights the need to understand the dynamic nature of the coupled ocean-atmosphere system in a changing climate for improving future predictions of regional precipitation.

[1] Department of Civil and Environmental Engineering, University of California, Irvine, CA 92697, USA. [2] Department of Earth System Science, University of California, Irvine, CA 92697, USA. Correspondence and requests for materials should be addressed to E.F.-G. (email: efi@uci.edu)

Seasonal and sub-seasonal precipitation prediction typically relies on deterministic climate models or statistical methods[1]. However, both climate models (which represent the complex land–ocean–atmosphere interactions) and statistical models (that either harness empirical relationships between precipitation and large-scale ocean–atmosphere teleconnections or are persistence-based) have shown limited success for precipitation prediction[1–7]. As climate change is expected to modify precipitation patterns, the need for improved seasonal to subseasonal predictive skill becomes critical for sustainable management of ecosystems and water resources[8–12].

Traditional drivers of winter precipitation in the southwestern US (SWUS) are the El Niño-Southern Oscillation (ENSO), decadal oscillations of the sea surface temperature (SST) in the Pacific and Atlantic oceans (i.e., Pacific Decadal Oscillation; PDO, Interdecadal Pacific Oscillation; IPO, Atlantic Multidecadal Oscillation; AMO), and persistent high-pressure ridges over the Gulf of Alaska[13–23]. Although these teleconnections are physically established[24–26], their predictive power on seasonal scales is limited. For example, there have been several years where El Niño conditions did not coincide with positive precipitation anomaly in SWUS, while years of extreme precipitation anomalies corre-

**Fig. 1** Evidence for a new teleconnection in the southwestern Pacific. **a**, **b** Correlation coefficients of –NZI and Niño 3.4 with winter precipitation (Nov–Mar) for all climate divisions of southwestern US (SWUS) and for two different periods (Jul–Sep and Sep–Nov) for 1950–2015. Average precipitation is defined as the area-weighted average precipitation amount over climate divisions where Niño 3.4 exhibits statistically significant correlation (see colored numbers in panels **a**, **b**, and regions in panel **h**). **c** Correlation map between SST (Jul–Sep) and the average winter precipitation in SWUS for 1950–2015. White color indicates statistically insignificant correlations ($\alpha = 0.05$ significance level); **d** Same as **c**, but SST is averaged over Sep–Nov; **e**, **f** Correlation maps as in **c**, **d**, but using GPH (400 mb). The emergence of a persistent correlation pattern in the southwestern Pacific (coined as the New Zealand Index, NZI) is robust for both SST and GPH; **g** the location and areal extent of NZI and Niño 3.4; **h** The selected climate divisions in SWUS (in color), for deriving the regionally averaged precipitation amount, based on their significant correlation with Niño 3.4

sponded to neutral ENSO conditions[20,22,27,28]. More generally, ENSO as well as other climate modes (e.g., PDO, AMO) exhibit weak statistical relationships with precipitation amounts, and even lower predictability as lead time increases[21,22,27,28].

In this study, we use historical records and reanalysis data to show that late-summer SST anomalies close to New Zealand strongly correlate with SWUS winter precipitation, outperforming all commonly used teleconnections, and enhancing the potential for earlier and more accurate prediction of precipitation. We provide evidence that the strength of the discovered teleconnection has been increasing during the latest 3–4 decades, in contrast to ENSO indices, which have been losing predictive strength. The new teleconnection is linked to an interhemispheric atmospheric bridge which occurs over western Pacific and is proposed to be driven by the migration of the intertropical convergence zone to the northern hemisphere during late summer, and the accompanied expansion of the southern Hadley cell[29], which allows for SST anomalies in the south to affect the north Pacific.

## Results

**Evidence for a new teleconnection.** Grounded on the hypothesis that still undiscovered relationships between large-scale atmosphere–ocean dynamics and SWUS precipitation might exist, we followed a diagnostic approach by which instead of restricting ourselves to the established teleconnections, we analyzed systematically the correlation of global SST and geopotential height (GPH) with winter precipitation amounts for all of the climatic divisions in SWUS. The premise was that if a coherent pattern emerged from such a data-analytic approach, it would

warrant merit for further investigation of its physical/mechanistic underpinning and its possible relation with other climate modes known to influence SWUS precipitation. For our analysis, we used observations of winter precipitation amount (data source: https://www.ncdc.noaa.gov/cag/time-series/us, [30]), together with SST data from two different datasets (monthly SSTs on a 1° × 1° grid, [31,32]) and GPH reanalysis data (monthly GPH on a 2° × 2° grid, [33]); see section Methods for more information. We assessed predictability based on global correlation maps between winter (Nov–Mar) precipitation amount in each climate division within SWUS (California, Nevada, Arizona, and Utah) and SST and GPH, averaged over 3-month periods corresponding to different lead times. We performed the analysis for the 66-yr period of 1950–2015, when higher quality SST and GPH observations are available. Specifically, for each grid cell on the globe, we calculated the correlation between $P_m$ and $I_{i:i+dt}^{k,l}$, where $P_m$ is the precipitation (Nov–Mar) in climate division $m$, $I_{i:i+dt}^{k,l}$ is the SST (or GPH) at latitude $k$ and longitude $l$, while $i$ indicates the starting month, and $i + dt$ the ending month of the period over which SST and GPH were averaged; here we considered $i = 7$ and 9 (Jul and Sep) and $dt = 2$, that is, we correlated $P_m$ (Nov–Mar) to $I$(Jul–Sep) and $I$(Sep–Nov), i.e., late summer and fall period. This analysis allowed us to examine the presence, and the strengthening or weakening of spatial correlation patterns with lead time.

The unexpected result of our investigation was the emergence of persistent SST and GPH patterns located in the southwestern Pacific, which exhibited strong negative correlation with precipitation in most SWUS climate regions. By examining the correlation maps for all climate divisions, the location and areal extent of the SST pattern was first qualitatively assessed to ensure its robustness. Then, by considering the SST anomalies enclosed

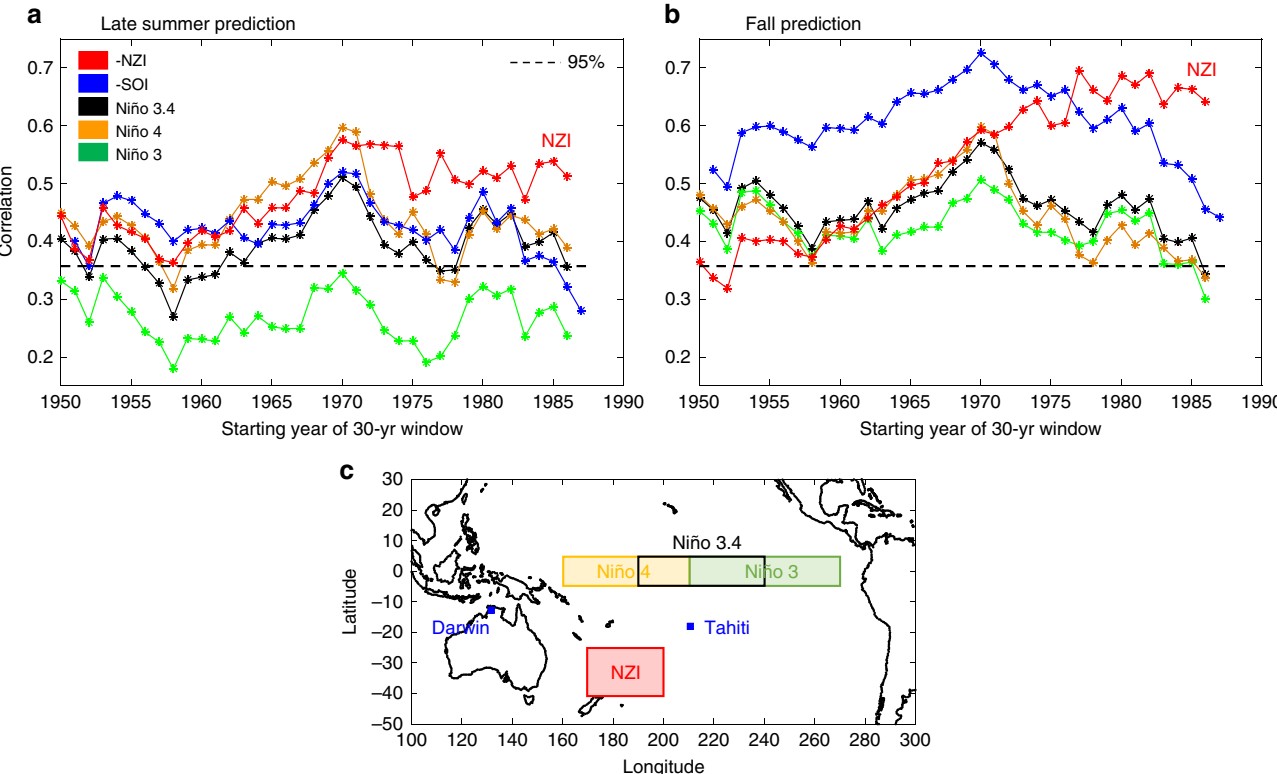

**Fig. 2** Strengthening of NZI after the mid-1970s. 30-year running averages of ENSO and NZI teleconnection strengths for two different lead times: **a** Correlation values for 30-yr moving windows between different climate indices (averaged over Jul–Sep) and winter precipitation in SWUS (Nov–Mar); **b** same as (**a**) but when climate indices are averaged over Sep–Nov. Dashed lines indicate the threshold below which correlations are statistically insignificant ($\alpha = 0.05$ significance level); **c** the location and areal extent of NZI and ENSO indices. SOI (Southern Oscillation Index[14]) is defined as the standardized difference of the sea level pressure anomalies at Tahiti and Darwin (Australia)

within contours of statistically significant correlation, the location and size of the emergent teleconnection was formally defined as the 30° × 15° region of 170 °E–200 °E and 25 °S–40 °S (Fig. 1g). We refer to the emergent teleconnection as the New Zealand Index (NZI; Fig. 1g) because of its proximity to New Zealand.

Correlations between NZI (Jul–Sep and Sep–Nov) and winter precipitation (Nov-Mar) for each climate division within CA, AR, UT, and NV revealed two important properties of the teleconnection (Fig. 1a, b). First, NZI correlations were stronger than Niño 3.4 correlations for most climate divisions, highlighting the potential for earlier and more accurate precipitation predictions than those afforded by ENSO. Second, correlations of NZI and ENSO exhibited similar variability among different regions, with precipitation amount over northern CA, NV, and UT showing lower correlation with both indices. This is not surprising, since precipitation in these regions is known to be less dependent on Pacific SSTs. Specifically, by modulating the latitude at which the jet stream prevails, Pacific SSTs influence the US climate over the northern and southern west coast, but not over regions in the central western US, which exhibit very different precipitation variability[13,14,16,23,34,35]. For the remainder of this study, we focused our analysis on the region where precipitation was significantly related to Pacific SST variability. We defined a regionally averaged SWUS precipitation time series as the area-weighted average precipitation amount over climate

divisions for which ENSO (Niño 3.4) exhibited a statistically significant correlation (at a significance level of $\alpha = 0.05$; Fig. 1a, b, h).

The spatial correlation patterns between SST and the area-weighted average SWUS precipitation are shown in Fig. 1c, d, for two lead times. Pacific SSTs in ENSO region had correlation values that did not exceed 0.4, whereas the strongest dependencies occurred in the southwestern Pacific, where correlations were less than −0.6. Similar patterns were apparent when using GPH data (400 mb), where a correlation of −0.5 was observed over the NZI region (Fig. 1e, f). The correlations with GPH did not depend on pressure level (Supplementary Fig. 1), revealing that the GPH pattern had a barotropic structure, a major characteristic of teleconnection patterns in the atmosphere[36].

**Decadal variation of precipitation teleconnections.** ENSO teleconnections to US precipitation have been shown to exhibit decadal and multidecadal variations in their strength[13–15,34,37,38]. These variations have been mainly attributed to natural climate variability[14,15,34] (e.g., PDO, AMO), and have been linked to the recurrence of persistent dry/wet periods in California and Arizona, with a periodicity of the order of 15 years[17,28] (Supplementary Fig. 2). Here, to explore the decadal variations and trends of NZI and ENSO teleconnection with winter precipitation in SWUS, we considered a 30-yr moving window (starting in

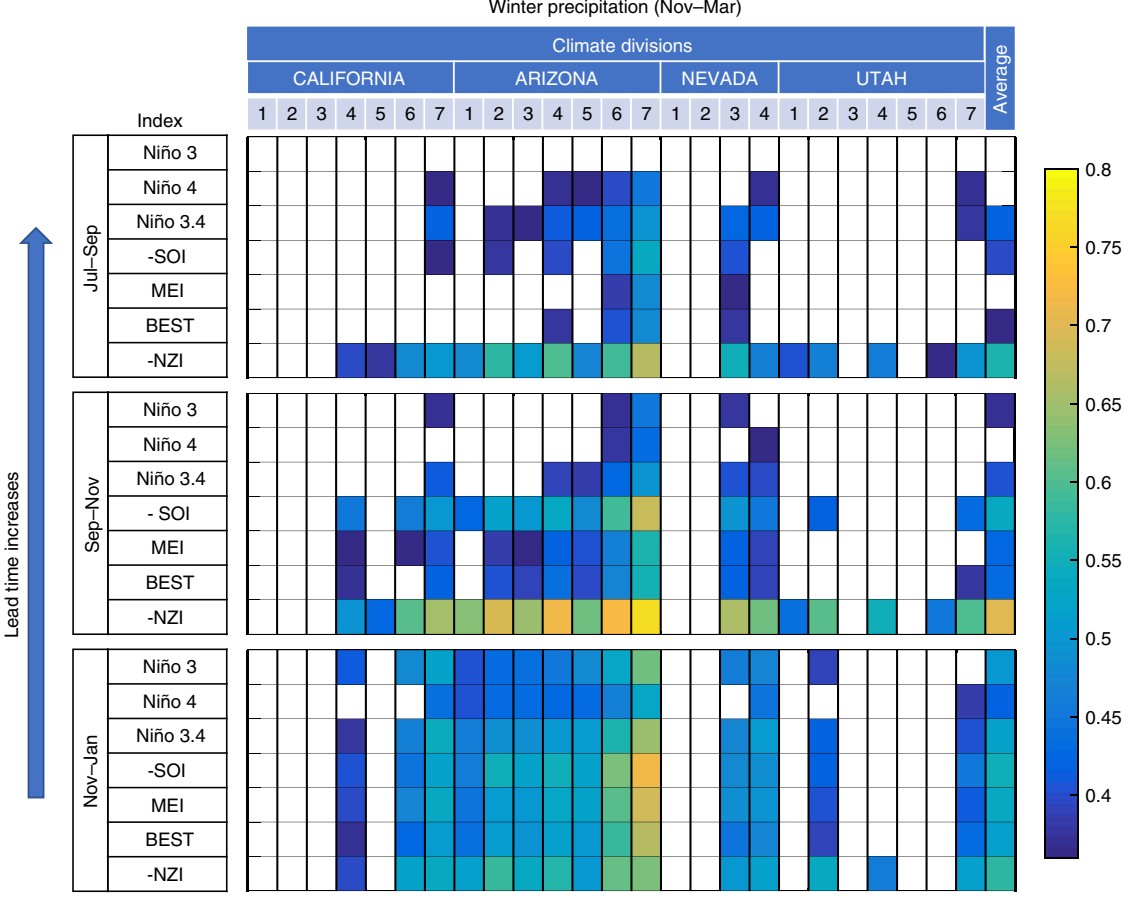

**Fig. 3** Comparison of ENSO and NZI teleconnection strengths. Colors illustrate correlation values between precipitation in different climate divisions over SWUS (Nov–Mar) and selected climatological indices corresponding to different 3-month periods, for 1982–2015. Colors denote statistically significant correlations ($\alpha = 0.05$ significance level). Results in the last column were calculated using the regionally averaged precipitation amount. NZI outperformed existing indices (correlation of −0.7 compared to 0.3–0.5) and provided statistically significant predictions with a 3-month lead time. Multivariate ENSO Index[5] (MEI) and Bivariate ENSO Timeseries[59] (BEST) are comprehensive indices consisting of multiple meteorological variables (SST, sea level pressure, surface wind, etc.) for determining ENSO state

1950), and calculated the correlation of NZI and ENSO indices averaged over periods of Jul–Sep and Sep–Nov with the regionally averaged winter precipitation amount. This analysis showed that the correlation of NZI with precipitation steadily increased over the duration of the time series, exceeding ENSO's correlation during the last 3–4 decades (Fig. 2a, b). ENSO indices, in contrast, showed a declining teleconnection strength since the 1970s.

Although the ENSO teleconnections had a similar decadal structure of variability over the record, differences among the indices have important implications for seasonal prediction. Niño 3 exhibited statistically insignificant correlation for the Jul-Sep interval (at $\alpha = 0.05$ significance level), indicating that eastern Pacific SSTs may not be effective for SWUS precipitation prediction. SOI, in contrast, had the strongest correlation with winter precipitation of all the ENSO indices (Fig. 2b), suggesting that atmospheric pressure variations across the tropical central/ western Pacific may be more effective for prediction than SST-derived indices. ENSO teleconnection strength did not seem to be affected by the increased frequency of central-Pacific ENSO events relative to those of the eastern-Pacific, which has been reported recently[39–41]. Specifically, SST-based ENSO indices for different zonal regions (including Niño 3, Niño 3.4, and Niño 4)

did not show diverging correlation patterns over the past several decades.

To quantitatively compare the predictive skill of NZI and ENSO indices, we sequentially performed cross-validation analysis (over 30-yr running windows) using three linear prediction models based on NZI, SOI, and Niño 3.4 time series. We assessed the predictive skill of each index based on the Root Mean Square Error (RMSE of the corresponding model prediction for regionally averaged winter (Nov–Mar) precipitation; the percentage of precipitation variability each model explained ($R^2$); and the probability of each model to correctly predict dry or wet conditions in the SWUS, with the former referring to precipitation amounts below the 33% quantile, and the latter above the 66% quantile. Based on all metrics, after the mid-1970s NZI's predictive skill increased, while ENSO indices lost predictability (see Supplementary Fig. 3). Specifically, in the past three decades, NZI's correlation to winter SWUS precipitation was about −0.7, while SSTs in the Niño 3.4 region exhibited almost statistically insignificant relationships (Supplementary Fig. 4). The correlation of NZI with winter precipitation for each climate division and for each of the different time lags is shown in Fig. 3, further establishing higher level of performance of NZI

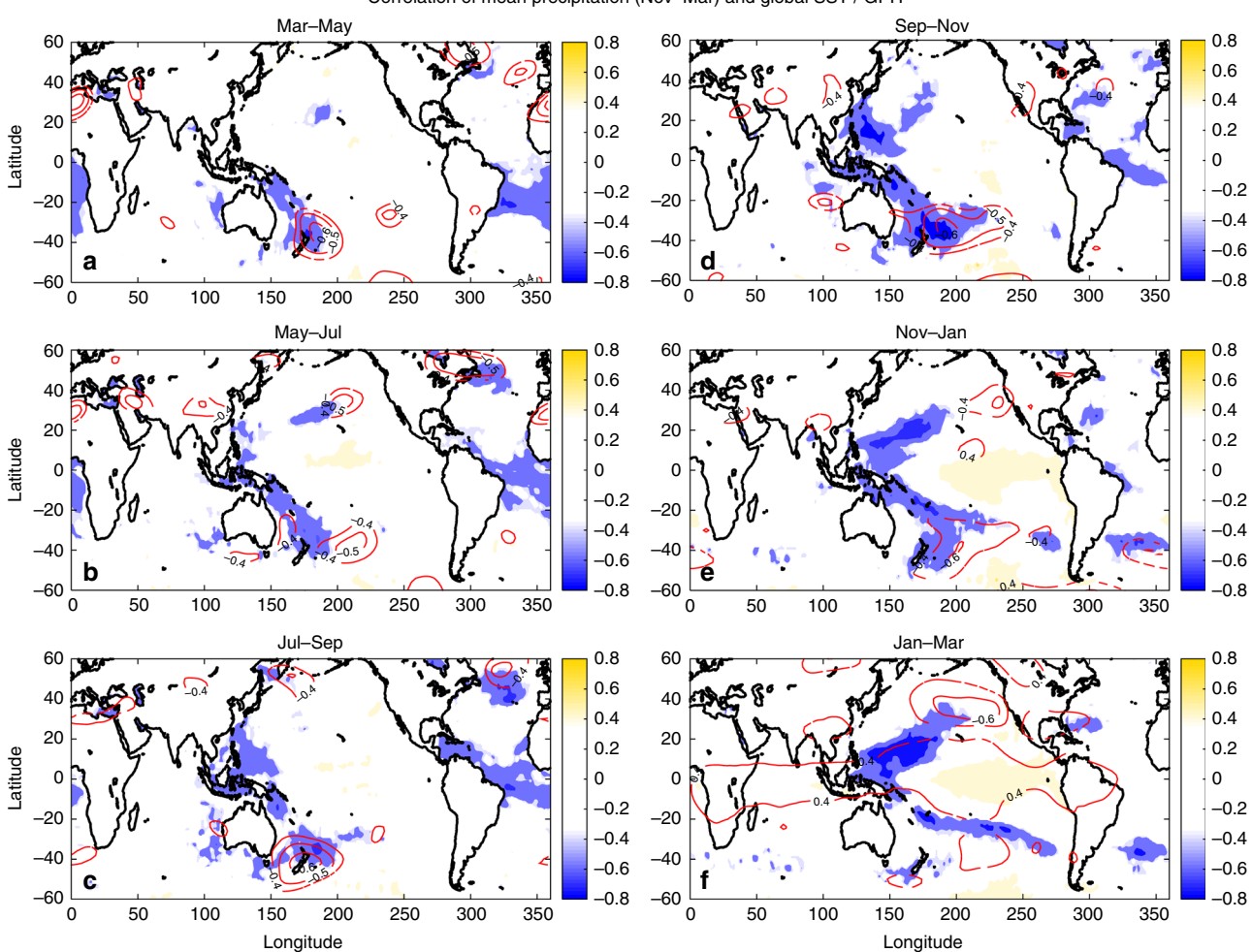

**Fig. 4** Within-year evolution of the association of winter precipitation in SWUS with SSTs and GPHs revealed a northward interhemispheric cascade. Correlation maps between average winter precipitation in SWUS (Nov–Mar) and global SST (shading) and GPH (400 mb; contours), for 1982–2015. SSTs and GPHs are averaged over different periods, starting Mar–May (previous spring) and ending Jan–Mar. Only statistically significant correlations are shown ($\alpha = 0.05$ significance level)

compared to the most commonly used ENSO indices. The correlation of winter precipitation and NZI in late summer (during Jul-Sep) was high (about $-0.55$), and increased more during Sep-Nov (about $-0.7$).

**The western Pacific pathway.** Important insight about the physical mechanism underpinning the NZI teleconnection is gained by exploring the dynamic evolution of the relationship between winter precipitation in the SWUS and ocean and atmosphere state variables in the Pacific basin during the preceding year. SSTs in the NZI region and SWUS precipitation exhibited statistically significant ($\alpha = 0.05$ significance level) correlations (Fig. 4) during boreal spring (correlations of $-0.5$), which strengthened during boreal summer, and reached maximum values in fall (correlations of $-0.7$). Approaching winter, the association of precipitation with the SST and GPH in the NZI region decreased significantly; see also the decrease in the correlation of NZI in the lower panel of Fig. 3. For zero lead time, precipitation was strongly correlated with SSTs in the northwestern Pacific and GPH directly to the west of SWUS[18–20,42,43], while correlations with ENSO regions emerged, but remained low (lower than 0.5). Two important conclusions were drawn from this analysis. First, climate information from the southwestern Pacific was critical for early prediction of the winter precipitation in the SWUS. Second, winter precipitation in SWUS was modulated by a northward cascade of SST and GPH anomalies through the year, starting in the southern hemisphere during late summer and fall, and cascading to the northwestern Pacific during fall and winter. We propose that this cascade was a key characteristic of the NZI teleconnection and worked through an atmospheric bridge in boreal summer, connecting SSTs in the southern hemisphere with those in the northern hemisphere. We present a mechanistic

explanation of the NZI teleconnection in Fig. 5, and elaborate on it below.

SST anomalies in the NZI region exhibited strong correlation with time-concurrent SST and GPH anomalies near the Philippines, with statistically significant correlations throughout the year (order of 0.5 or less), which intensified during late boreal summer reaching values of 0.7 or higher (Supplementary Fig. 5). By analyzing the zonal average vertical velocity over the eastern Asia—western Pacific region of 70 °E–220 °E (Supplementary Fig. 6), we found that positive SST anomalies in the NZI region induced an anomalous Hadley circulation that ascended from the southwestern Pacific and descended over the Philippines. Its descending motion produced adiabatic warming and increased downward shortwave radiation (likely by means of suppressing cloud formation) resulting in positive SST anomalies near the Philippines. The descending motion also produced positive GPH anomalies. This inter-hemispheric teleconnection was strongest during boreal summer (Supplementary Fig. 5), when the southern Hadley cell expands the most[29], connecting the two regions (Supplementary Fig. 7). The maximum correlation between late summer NZI and SSTs in the northern hemisphere occurred during late fall (see Supplementary Fig. 8) in the eastern side of the Philippines (as high as 0.85). This 3–4 months lag is a main characteristic of the atmospheric bridge and is associated with the time needed for ocean surface heat content and SSTs to respond to the cumulative atmospheric forcing[44]. A similar inter-hemispheric relation, connecting 500 mb GPH anomalies in the area east of Australia with those over the Philippines has been reported and described in the literature[45–47]. Specifically, past studies document that the latter teleconnection also takes place during late summer, and that the longitudinal zone around the globe where the most significant interhemispheric interaction occurs is the region of east Australia to east Asia[45,46].

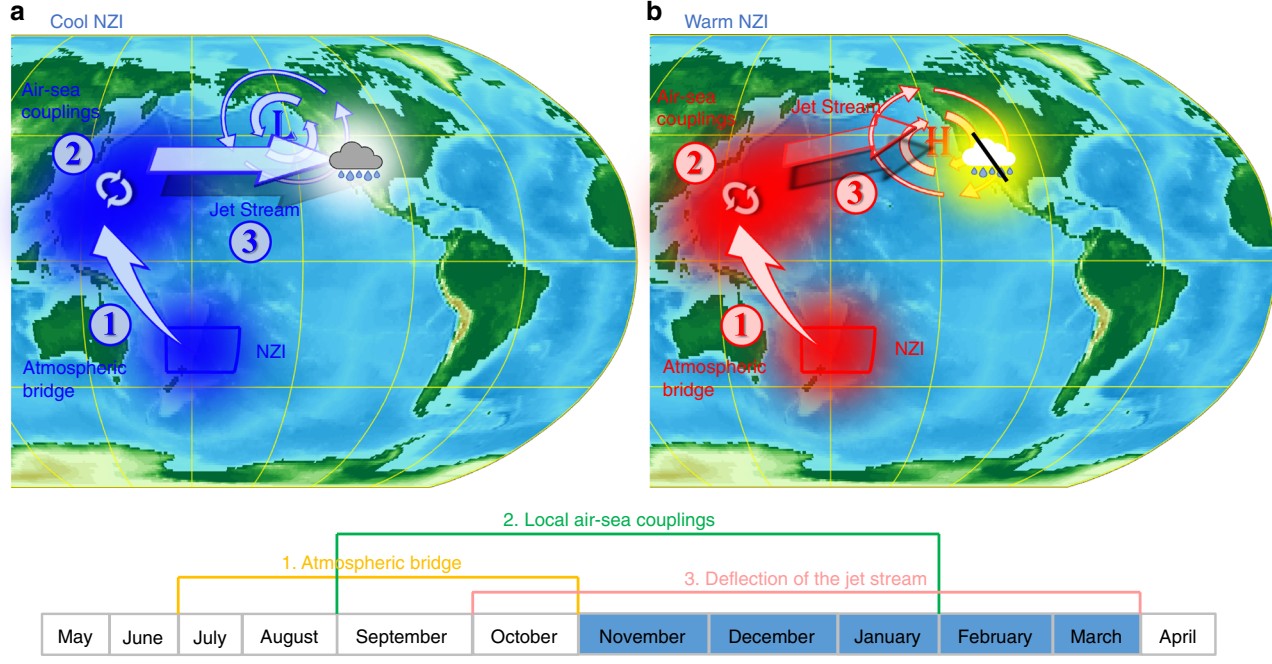

**Fig. 5** The New Zealand Index (NZI) teleconnection depends on a western Pacific ocean–atmosphere pathway. **a** Negative SST anomalies (blue shading) in the NZI region cascade in the northern hemisphere through a late summer interhemispheric atmospheric bridge and are maintained by air-sea coupling until the following winter. The SST anomalies affect the atmospheric pressure in the US west coast and strengthen the regional jet stream which brings more winter storms in the SWUS; **b** Late-summer positive SST anomalies (red shading) in the NZI region deflect the jet stream to the north, leading to dry conditions over the SWUS

As soon as the cascade to the northern hemisphere has occurred, SST anomalies in the northwestern Pacific persist through boreal winter (see Supplementary Fig. 8) via local atmosphere–ocean couplings. Previous studies have shown that SST anomalies in this region interact with local mean trade winds to induce anomalous surface heat flux, and create a positive feedback loop between anticyclonic (cyclonic) activity and sea surface cooling (warming)[48]. This mechanism allows for maintenance and propagation of the SST anomalies from the northwestern Pacific to the northcentral Pacific (Supplementary Fig. 8). When the SST anomalies in the western and central north Pacific persist into the boreal winter, they alter the atmospheric pressure in the western US through a short-wave train which links the two areas[20,49], modifying the track and strength of the jet stream over the west US coast (220 °E–260 °E). These changes in the jet stream can steer more or fewer winter storms to impact SWUS precipitation (see Figs. 4e, f and 5). Specifically, cool NZI conditions during late summer, after propagating into the northwestern Pacific and maintaining themselves during fall and winter, intensify the upper zonal winds at 30–35 °N, and bring above average precipitation to SWUS (Fig. 6). In contrast, warm SSTs in the NZI region excite positive anomalies in the northwestern Pacific, deflecting the jet stream northward (at 45°–50 °N), and leading to wet conditions over the northwestern US, and dry conditions in SWUS. Accordingly, NZI is associated with a dipole pattern along the west coast of the US, which is evident in both zonal winds of the upper atmosphere, as well as in precipitation amounts (see Supplementary Fig. 9). Note that this behavior is well-known to be also associated with ENSO events[13,14,16,23,34,35], yet in the last three to four decades, ENSO did not correlate significantly with precipitation amount for many regions along the west coast (Supplementary Fig. 10).

The proposed mechanism of NZI reveals a western Pacific pathway and can be summarized by the following three sub-processes (Fig. 5). (1) In late summer, expansion of the Southern Hadley cell creates an interhemispheric atmospheric bridge connecting geopotential heights (and SSTs) in the area close to New Zealand with those in the northwestern Pacific. Similar interhemispheric teleconnections have been described in previous studies[45–47]. (2) In late fall, local air-sea coupling maintains atmospheric pressure and SST anomalies in the northwestern Pacific[48]. (3) Finally, in winter SSTs in the northwestern Pacific influence atmospheric pressure and upper zonal winds in the western US, which in turn change the positioning of storm tracks and precipitation amount in SWUS[18–20,42,43,49].

**Changes in pacific dynamics.** By establishing similar analyses in the period before the 1980s, we determined that the correlation of NZI with SSTs in the region east of the Philippines (referred to as EPh: the areal average SSTs in the region of 5–15 °N and 130–150 °E), has not been stationary (Fig. 7). During 1950–1983, the connection of NZI and EPh was weak (correlations did not exceed 0.45), whereas during the last 35 years, it was remarkably strong and robust, with correlations reaching 0.8 during boreal fall, and with NZI leading EPh by 3–4 months. The intensification of this connection between the two hemispheres was the primary reason for the increased correlation of NZI and SWUS precipitation observed during the last several decades (Fig. 2). In contrast, during the earlier period where the NZI was only weakly connected to the northern hemisphere, its correlation with precipitation variability in SWUS was lower.

The time-evolving covariance of SSTs in the two regions highlights the non-stationary nature of ocean-atmosphere dynamics on decadal time scales, and its direct impact on regional hydroclimatic variability. The intensification of this

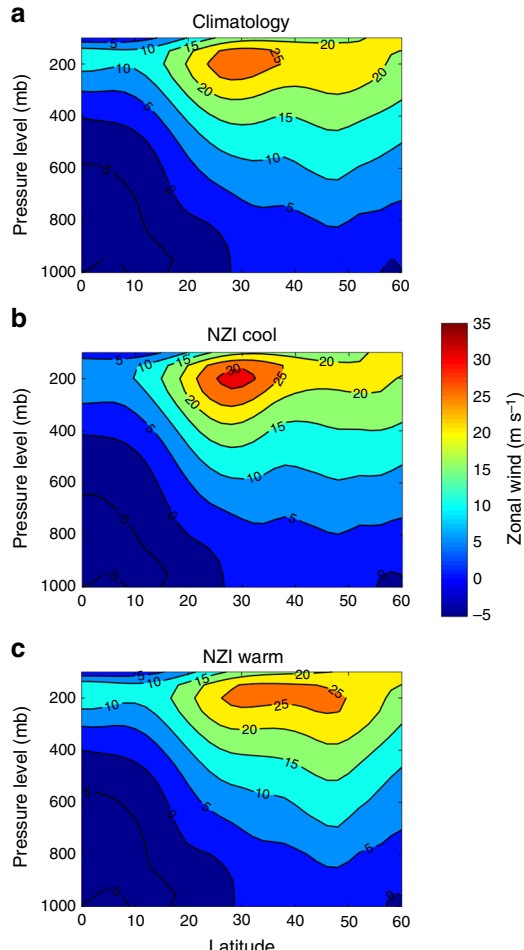

**Fig. 6** NZI affects the upper zonal winds in the northeastern Pacific and modulates SWUS precipitation. **a** Zonal average (220°E–260°E) of zonal wind in m s⁻¹ for different latitude and pressure levels, during Nov–Mar, for 1982–2015; **b** Same as (**a**), but for the 5 coolest NZI years; see Supplementary Fig. 4; **c** Same as (**a**), but for the 5 warmest NZI years

northward interhemispheric cascade may be a result of the expansion of the tropics which has been observed after the 1980s[11,50–53,54], and allows SST or GPH variability at lower latitudes to more strongly affect the meridional atmospheric circulation at higher latitudes. However, it is still not known what exactly is the relative contribution of the natural variability (multi-decadal oscillations) and external (anthropogenic) forcing to the latter change[52,53], so it is not certain whether the strength of the NZI signal will continue to increase or undergo periodic fluctuations in the future. The observed increase of the correlation of NZI with precipitation may also be influenced by changes in data quality after the late 1970s, when satellite data improved coverage, particularly in the southern hemisphere[24,55].

## Discussion

Non-stationary relationships between potential predictors (ENSO or NZI) and precipitation due to climate change or internal climate variability can significantly impact our ability to develop accurate seasonal forecasts[14,56,57]. The revealed changes in the strength of the NZI and ENSO signals to winter precipitation in SWUS is an example of such variations, offering the chance to quantify the effect of large-scale non-stationarity on regional precipitation predictability. To illustrate this point, we evaluated and compared the prediction error of an SST-based regression

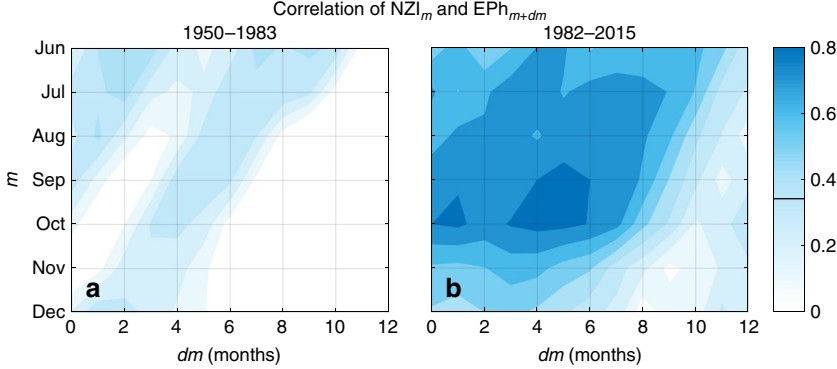

**Fig. 7** Interhemispheric teleconnections between NZI and SST anomalies in the northwestern Pacific have significantly strengthened during the last three to four decades. **a** Time-lagged correlation between NZI (averaged over the month indicated in the vertical axis) and SST in the region east of the Philippines (EPh) (box in 5–15°N and 130–150°E; lagged forward in time as indicated in the horizontal axis) in 1950–1983. Solid black line on the colorbar indicates the threshold below which correlations are statistically insignificant ($\alpha = 0.05$ significance level); **b** Same as (**a**), but for the period 1982–2015

model in settings in which the calibration and prediction periods coincided vs. settings that they did not (i.e., in-sample vs. out-of-sample predictions). Specifically, we predicted winter precipitation over two non-overlapping periods (1951–1983 and 1983–2015) using a multiple regression model based on NZI, SOI and Niño 3.4 series. We quantified the effect of non-stationarity on the prediction skill as the relative increase of RMSE when the calibration and prediction periods did not overlap (e.g., prediction in 1983–2015, calibration in 1951–1983) relatively to the case that the calibration and prediction periods completely overlapped (e.g., prediction in 1983–2015, calibration in 1983–2015). If stationary conditions held, the increase of RMSE would be close to zero, since calibration in either period would result to almost identical regression functions (deviation from zero would be a result of sample variability). However, this analysis revealed that the relative increase of RMSEs can be as high as 400–500% (Supplementary Table 1), which means that our predictive skill is ~5–6 times lower than what it would have been if stationary relationships actually held. The latter analysis demonstrates the important effect of the stationarity assumption in a changing climate, and highlights the importance of understanding the multi-scale patterns of change from climate dynamics to regional precipitation.

In summary, we show that during the last several decades, SST anomalies in the southwestern Pacific Ocean have been closely related to precipitation in SWUS through a western Pacific pathway, and thus, they may be useful in increasing precipitation predictability. Idealized climate model experiments can be used to provide further evidence for our proposed teleconnection mechanism, by quantifying how prescribed SST anomalies in the southwestern Pacific influence the meridional circulation, north Pacific SSTs, and the western US precipitation. Further research is also needed to better understand the drivers of the decadal variability of the newly established teleconnection. Our work emphasizes the need to understand the dynamic nature of the coupled atmosphere–ocean system in a changing climate, for improving future predictions of regional precipitation.

## Methods

**Correlation significance.** For estimating the 95% intervals corresponding to statistically insignificant linear correlation, we assume a $t$-distribution[52]: $r_c = \frac{\pm t}{\sqrt{N-2+t^2}}$, where $t$ is the 97.5% quantile of the $t$-distribution, with d.f. $=N-2$, and $N$ is the sample size.

**Correctly predicting wet and dry conditions.** The probability of wet hit ($P_w$) used in Supplementary Fig. 3e, f is defined as

$$P_w = Pr[\hat{P} > P_{66\%} | P > P_{66\%}] = \frac{Pr[\hat{P} > P_{66\%} \cap P > P_{66\%}]}{Pr[P > P_{66\%}]} = \frac{I[\hat{P} > P_{66\%} \cap P > P_{66\%}]/N}{I[P > P_{66\%}]/N},$$

where $\hat{P}$ and $P$ is the predicted and observed precipitation, respectively, $P_{66\%}$ is the 66% precipitation quantile (based on the entire record), $I[\cdot]$ is the indicator function, and $N$ is the sample size (in Supplementary Fig. 3, N = 30). The probability of dry hit ($P_d$) is defined in a similar way as $P_d = Pr[\hat{P} < P_{33\%} | P < P_{33\%}]$.

**Data availability.** All data used in this study are freely available. Observations of winter precipitation amount in all climate divisions over the US region are freely available at https://www.ncdc.noaa.gov/cag/time-series/us, [30]. Global precipitation records (see Supplementary Figs. 7 and 9b) are obtained from http://chrsdata.eng.uci.edu/ (monthly precipitation on a 0.25° × 0.25° grid, [58]). SST data were obtained from https://www.esrl.noaa.gov/psd/data/gridded/data.cobe2.html (monthly SST on a 1° × 1° grid, [31]) while GPH, zonal, meridional and omega velocities were obtained from https://www.esrl.noaa.gov/psd/data/gridded/data.20thC_ReanV2c.pressure.mm.html (monthly series on a 2° × 2° grid, [33]). Results in Figs. 3 and 4 and Supplementary Figs. 4–6, 8–10 are derived based on an alternative SST dataset freely available at https://www.esrl.noaa.gov/psd/data/gridded/data.noaa.oisst.v2.html (monthly SST on a 1° × 1° grid; [32]), to ensure robustness of our analysis and conclusions, using different reanalysis datasets. Time series of ENSO indices (monthly scale) were obtained from https://www.esrl.noaa.gov/psd/data/climateindices/list. Also, upon reasonable request, the data and code that support the findings of this study can be provided by the corresponding author.

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

## Acknowledgement

The authors would like to thank three anonymous reviewers for their helpful comments and suggestions that greatly improved the manuscript, especially in strengthening the mechanistic explanation of the new teleconnection. Partial support for this research was provided to E.F.-G by the National Science Foundation (NSF) under the Water Sustainability and Climate Program (grant EAR-1209402) and the Science Across Virtual Institutes Project (LIFE: Linked Institutions for Future Earth, grant EAR-1242458), and by NASA's Global Precipitation Measurement (GPM) Program (grant NNX16AO56G). J.-Y.Y. was supported by NSF under the Climate and Large-Scale Dynamics Program (grant AGS-1505145). J.T.R. was supported by the Gordon and Betty Moore Foundation (GBMF3268) and the NASA's SMAP Program. A.A. was supported by the National Oceanic and Atmospheric Administration (NOAA) award NA14OAR4310222 and the NASA award NNX15AC27G. A research grant from UCI to all PIs to advance these research ideas is also acknowledged.

## Author contributions

A.M. designed the study, performed the data analysis, and wrote the manuscript. E.F.-G. supervised the research, methods of analysis and interpretation, and assisted in the manuscript writing. J.-Y.Y. and J.T.R. helped with the design of several analyses, and contributed to the physical interpretation of the results and to the manuscript writing.

A.A. assisted with data analysis. All authors contributed to the interpretation of the results.

## Additional information

**Competing interests:** The authors declare no competing interests.

