## [Peer Review File · Nature Communications]

Reviewers' comments:

Reviewer #1 (Remarks to the Author):

The paper claims that late-summer persistent sea surface temperature (SST) and geopotential height anomalies (GPA) in the southwestern Pacific exhibit higher correlation with southwestern US (SWUS) winter precipitation than the El Niño-Southern Oscillation (ENSO). Additionally, the authors suggest an amplification of this new teleconnection represented by the so-called New Zealand index (NZI) has occurred over the past four decades, and most likely is due to the expansion of the tropics. The most interesting aspect is perhaps the apparent strengthening of the correlation between SWUS precipitation and NZI, an amplification that is not observed in the ENSO teleconnection.

For reasons expressed in the overall comments, however, I cannot recommend acceptance of this paper at its present form.

Overall comments: If we step back and look at the manuscript as a whole, then basically what it's doing is some correlation of time series to discern various phase relationships between the time series. While this approach is fine as supplementary, supporting information to a physical argument, in this manuscript there is too little physical argument for why these phase relationships should be there. For example, there is well-developed physical theory for why ENSO directly modifies precipitation over western U.S. By comparison, why, physically, should precipitation in this region be modulated by SST changes in the other hemisphere, as claimed here?

The following figure [redacted] from Hoskins and Woollings (2015) displays the major "routes" of atmospheric teleconnection, comprising both long (zonal) and short (poleward) Rossby wave propagations. While newer modes of atmospheric teleconnection (other than ENSO-induced ones) have been found, the primary teleconnection routes remain unchanged. As this figure shows, these teleconnection routes do not include anything that crosses the hemispheres from New Zealand to SW US, as claimed by the authors.

From my understanding of the manuscript, there are two conflicting proposals concerning the cause-and-effect relationship: (1) that NZI modulates SWUS directly (inferred from Fig.1&2) or (2) that NZI leads ENSO and then ENSO modulates SWUS (inferred from Fig. 4). However, for (1), as noted in the left figure, there is no such route as an NZI-North America teleconnection existed and this paper has not provided any reason why there should be one. For (2), the ENSO correlation with the SWUS precipitation has declined as the authors noted (though it remains significant) and so, physically, it should not explain the increasing correlation of NZI-precipitation through ENSO.

The "Changing Pacific Dynamics" section of the manuscript is well written, but I found it not so relevant to the NZI-SWUS/P argument and somewhat distracting. The IPO/PDO modulation on ENSO and its teleconnection to SWUS has been proposed with both empirical and modeling evidence. Using similar arguments to explain the unexplained NZI-SWUS (tele)connection is premature.

Specific Comments:

1: The authors write - "here we report evidence that late-summer persistent sea surface temperature and geopotential height anomalies in the southwestern Pacific exhibit higher correlation with SWUS winter precipitation than ENSO." I feel this is not necessarily true since the correlation analysis is done using average precipitation corresponding to climate divisions where

Niño 3.4 exhibits statistically significant correlation. More so, the NZI teleconnection strength (Figure 3) varies considerably for different climate divisions. I suggest making the distinction clearer in the text.

2: "Notably, our analysis shows that the GPH results do not depend on the considered pressure level, further indicating the robustness of NZI and providing confidence on its physical underpinning. The similar correlations at various pressure levels also reveal that the GPH pattern has a barotropic structure"- I don't think this section in its entirety is evident from any of the figures. Where is the analyses for multiple pressure levels shown?

3: It would be interesting to see the lagged correlations between the NZI/ENSO indices and SWUS precipitation, with increasing lead times (perhaps starting a year prior) rather than averaging climate indices over the preceding three to six months (i.e. just Jul-Sep and Sep-Nov) and then presenting the best results.

4: Although the NZI yields relatively higher correlations with SWUS precipitation (Figure 1), the results from Figure S2 are not statistically robust enough to suggest the NZI can be utilized for useful predictions in a way that ENSO indices may not. Although a more compelling argument can be made for the period following 1970 (Figure 2b), specifically for fall; but I believe this will merit further research, including analysis o

5: The authors discuss the decadal variation of precipitation teleconnections and specifically point to a "periodicity of the order of 15 years," but then elect to show a moving correlation analyses using only a 30-yr moving window. Could the authors comment on why they chose the 30-yr window? It would help to show the moving correlation across different windows (maybe ranging from 10-30 with 5 years' increments).

6: Given the scarcity of data observation networks in the southern Hemisphere in comparison to the North, it will be of good value to add multiple reanalysis datasets for the most prominent analyses to serve as verification.

Reviewer #2 (Remarks to the Author):

The subject manuscript is interesting and well written, and identifying climate teleconnections beyond the established teleconnections is important and useful. Although the paper has potential to be a useful contribution to the literature there are several issues that need to be addressed before the paper is suitable for publication. There are still several analyses that are needed to convincingly show the importance and reliability of the NZI.

Specific comments and suggestions:

1. The authors should consider referencing

Newman, M., S.-I. Shin, and M. A. Alexander, 2011b: Natural variation in ENSO flavors. *Geophys. Res. Lett.*, 38, L14705, doi:10.1029/2011GL047658.

Johnson, N.C., 2013, How many ENSO flavors can we distinguish? *Journal of Climate*, 26:4816-4827.

2. Redmond and Koch (1991) performed one of the first studies of the predictive ability of ENSO indices. The authors should reference this highly cited paper –

Redmond, K. T., and R. W. Koch, 1991: Surface climate and streamflow variability in the western

United States and their relationship to large-scale circulation indices. *Water Resour. Res.*, 27, 2381–2399.

3. The authors should provide an explanation for the selection of the 1950 through 2015 for analysis. Climate division data are available since 1895 and SST data go back to 1856. Why was the selected period chosen for this study?

4. It would be helpful and interesting to compare the spectral frequencies of precipitation in the southwestern U.S. with the spectral frequencies of ENSO and NZI. What are the common spectral frequencies and do they change during the period analyzed?

5. Why was the region of the US used for the analysis restricted to the climate divisions with significant ENSO correlations? Also, why not look at precipitation correlations for the entire western U.S.? Does NZI result in a western U.S. precipitation dipole similar to ENSO?

6. A possible useful twist to this analysis would be to perform a couple of different analyses – (1) perform a principal components analysis of western U.S. precipitation and correlate the significant PC score time series with SSTs and see if the NZI index shows up in one of the correlation fields? (2) perform a spectral analysis of western U.S. precipitation and then correlate time series of significant spectral frequencies with SSTs and see if the NZI shows up.

7. Something that would really strengthen this paper is some analysis and explanation describing how the NZI affects atmospheric circulation and results in an effect on precipitation in the western U.S. How do changes in the southwestern Pacific move across the tropics and influence the mid-latitude atmospheric circulation of the Northern Hemisphere?

8. Do long-term trends in NZI influence the results? There is a long-term warming trend in the NZI that is not seen in NINO3.4 SSTs and this warming trend may be influencing the results. The authors should consider removing long-term trends in all data sets and re-computing the results.

Reviewer #3 (Remarks to the Author):

the authors present a not-well-known teleconnection and even less well known predictive measure for seasonal precipitation in the Southwest U.S. I believe the results will be useful to researchers and forecasters. Methods and analyses are appropriate, a few suggestions/questions appear below.

The fall NZI to winter SW precipitation relationship would be more meaningful if you were able to show how anomalous atmospheric circulation that actually produces above normal precipitation in the Southwest US evolves from a distant, un-connected SST and GPH anomaly over New Zealand.

a few comments regarding the ms.

"72 SWUS, which is defined as the area-weighted average precipitation amount over climate divisions for which ENSO (Niño 3.4) exhibits statistically significant correlations (significance level $\alpha = 0.05$)" not standard definition of SW U.S. Explain why the ENSO connected region is adopted for purpose of testing NZI connection.

"86 precipitation are shown in Figs 1c-d, for two lead times: Jul-Sept and Sept-Nov" lead times usually refer to the time lag between predictor and predictand. the Sept-Nov predictor is problematic in context of prediction because it overlaps w the Nov-Mar precipitation predictand.

"90 the GPH results do not depend on the considered pressure level, further indicating the robustness of NZI and providing confidence on its physical underpinning." it is not unusual (actually expected) that multiple pressure levels vary coherently; multiple levels being involved does not indicate a robust association -- one being correlated essentially brings the others along.

"198 We quantified the effect of non-stationarity on the prediction skill as the relative increase of RMSE when the calibration and prediction periods did not overlap (e.g. prediction in 1983-2015, calibration in 1951-1983) relatively to the case that the calibration-prediction periods completely overlapped (e.g. prediction in 1983-2015, calibration in 1983-2015). " it is difficult to understand this; producing two separate correlations is a pretty meager cross validation.

relevant references and findings that are not cited --

Cayan et al 1998 described. at decadal time scales, the association of SST and circulation in the western North and South Pacific, (including the New Zealand area) with anomalous precipitation in the SW U.S. Dettinger et al. 1998 show both decadal and interannual trans-Pacific SST and sea level pressure patterns that accompany anomalous precipitation in western N. America, including a distinct SW US pattern.

Cayan, D.R., Dettinger, M.D., Diaz, H.F., and Graham, N., 1998, Decadal variability of precipitation over western North America: *Journal of Climate*, 11, 3148-3166.

Dettinger, M.D., Cayan, D.R., Diaz, H.F., and Meko, D., 1998, North-south precipitation patterns in western North America on interannual-to-decadal time scales: *Journal of Climate*, 11, 3095-3111.

Reviewer #1 (Remarks to the Author):

General Comment 1

The paper claims that late-summer persistent sea surface temperature (SST) and geopotential height anomalies (GPA) in the southwestern Pacific exhibit higher correlation with southwestern US (SWUS) winter precipitation than the El Niño-Southern Oscillation (ENSO). Additionally, the authors suggest an amplification of this new teleconnection represented by the so-called New Zealand index (NZI) has occurred over the past four decades, and most likely is due to the expansion of the tropics. The most interesting aspect is perhaps the apparent strengthening of the correlation between SWUS precipitation and NZI, an amplification that is not observed in the ENSO teleconnection.

For reasons expressed in the overall comments, however, I cannot recommend acceptance of this paper at its present form.

We thank the Reviewer for his/her critical and careful consideration of our work. We have performed extensive new analysis documenting the physical underpinning of the NZI teleconnection and its recent intensification. Point by point responses are provided below.

General Comment 2

If we step back and look at the manuscript as a whole, then basically what it's doing is some correlation of time series to discern various phase relationships between the time series. While this approach is fine as supplementary, supporting information to a physical argument, in this manuscript there is too little physical argument for why these phase relationships should be there. For example, there is well developed physical theory for why ENSO directly modifies precipitation over western U.S. By comparison, why, physically, should precipitation in this region be modulated by SST changes in the other hemisphere, as claimed here?

[Redacted]

The following figure from Hoskins and Woollings (2015) displays the major “routes” of atmospheric teleconnection, comprising both long (zonal) and short (poleward) Rossby wave propagations. While newer modes of atmospheric teleconnection (other than ENSO-induced ones) have been found, the primary teleconnection routes remain unchanged. As this figure shows, these teleconnection routes do not include anything that crosses the hemispheres from New Zealand to SW US, as claimed by the authors.

*Thank you for the critical comment. Yes, it is indeed the interhemispheric nature of the NZI teleconnection that makes it new and non-trivial. The new section **The physical basis for the NZI teleconnection – The western Pacific pathway**, accompanied with additional 10 figures (4 in the main text and 6 in the supplementary material) present details on the new analysis and results. Below we respond to each comment separately.*

General Comment 3

From my understanding of the manuscript, there are two conflicting proposals concerning the cause-and-effect relationship: (1) that NZI modulates SWUS directly (inferred from Fig.1&2) or (2) that NZI leads ENSO and then ENSO modulates SWUS (inferred from Fig. 4). However, for (1), as noted in the left figure, there is no such route as an NZI-North America teleconnection existed and this paper has not provided any reason why there should be one. For (2), the ENSO correlation with the SWUS precipitation has declined as the authors noted (though it remains significant) and so, physically, it should not explain the increasing correlation of NZI-precipitation through ENSO.

We thank the Reviewer for his/her comment.

Concerning (1):

*We have now provided extensive physical arguments and new quantitative analysis as to the mechanism of NZI and its direct relevance to SWUS precipitation – see new section **The physical basis for the NZI teleconnection – The western Pacific pathway**, accompanied by additional figures and results. The summary of the NZI physical mechanism is depicted in new Fig. 5 which was designed to tell the story in a single graph, with details documented in further figures.*

Concerning (2):

We do not suggest that the connection of NZI to precipitation in SWUS occurs through the ENSO teleconnection. Actually, our argument is that the discovered teleconnection is separate from ENSO, and has been increasing in strength and decoupling from ENSO dynamics, especially during the last 3-4 decades.

The “Changing Pacific Dynamics” section (also Figure 4) of the previous version of the manuscript aimed to highlight this decoupling and suggest a propagation of NZI SST anomalies to the north Pacific; not specifically to the ENSO or the IPO region. Supplementary figure S5 and the GIF file supported this cascade.

*However, we understand that the presentation of our arguments might not have been as clear as intended. For that reason we do not include now old Fig. 4 in the revised version. Instead we have from scratch written section **The physical basis for the NZI teleconnection – The western Pacific pathway** to explicitly explain the NZI teleconnection and show how it relates to an interhemispheric bridge during boreal summer, local air-sea couplings that maintain atmospheric pressure and SST anomalies in the northwestern Pacific in late fall, and a jet-stream influence on storm tracks that modulates SWUS precipitation in winter. Please see the new section and new Figs in the main text (Fig.4-7) and the supplementary material (Figs. S5-S10) for details.*

General Comment 4

The “Changing Pacific Dynamics” section of the manuscript is well written, but I found it not so relevant to the NZI-SWUS/P argument and somewhat distracting. The IPO/PDO modulation on ENSO and its teleconnection to SWUS has been proposed with both empirical and modeling evidence. Using similar arguments to explain the unexplained NZI-SWUS (tele)connection is premature.

The “Changing Pacific Dynamics” section (also Figure 4) of the previous version of the manuscript aimed to highlight this decoupling and suggest a propagation of NZI SST anomalies to the north Pacific; not specifically to the ENSO or the IPO region. Supplementary figure S5 and the GIF file supported this cascade.

However, we understand that the presentation of our arguments might not have been as clear as intended and we have added new analysis and discussion on this point – please see the detailed response to your general comment 3.

Specific Comment 1

The authors write - “here we report evidence that late-summer persistent sea surface temperature and geopotential height anomalies in the southwestern Pacific exhibit higher correlation with SWUS winter precipitation than ENSO.” I feel this is not necessarily true since the correlation analysis is done using average precipitation corresponding to climate divisions where Niño 3.4 exhibits statistically significant correlation. More so, the NZI teleconnection strength (Figure 3) varies considerably for different climate divisions. I suggest making the distinction clearer in the text.

We thank the Reviewer for his/her comment.

NZI is shown to be more correlated with precipitation than ENSO in almost all cl. divisions (see results in Figs 1 and 3). Also, based on Fig. 1, Niño 3.4 is shown to exhibit quite variable connection strengths in different cl. divisions, similarly to NZI. This is consistent with the literature, since ENSO signal to precipitation in the northern side of California, Nevada and Utah is significantly weaker than that in the south¹⁻⁵. Note that Fig. 3 does not facilitate obtaining the latter variability of ENSO indices, since most ENSO correlation values in Fig. 3 are not statistically significant (especially for nonzero lead time) and are represented in white color, thus the variability is hidden (whereas in Fig. 1, one can see all correlation values for all climate divisions). To make the latter clear, we have added a statement in the manuscript (see lines 86-96).

Concerning the definition of the area-mean precipitation:

Precipitation amount over climate divisions which are located in the northern CA, NV and UT shows insignificant correlation with both NZI and ENSO indices. Indeed, precipitation in the northern side is known not to be related to Pacific SSTs, and to exhibit very different variability than in the rest of the SWUS see Figs 1, 3, S10 and [1-5]. To account for this natural disassociation of the northern region, we define the regionally averaged SWUS precipitation as the area-weighted average precipitation amount over climate divisions for which ENSO (Niño 3.4) exhibits statistically significant correlations (significance level $\alpha = 0.05$; Figs 1a-b and 1h). Notice that defining the average precipitation based on the climatic divisions with significant NZI-correlations does not affect the results (see figure below); moreover, working with the ENSO-

correlated sites as we did, places inferences related to the NZI teleconnection strength on the conservative side. We have added a discussion in the manuscript; see lines 86-96.

Figure 1. Same results as in Figs 1c-f of the manuscript, but the regionally averaged precipitation amount is calculated over climate divisions where NZI exhibits statistically significant correlation (see colored regions in panel e).

Specific Comment 2

“Notably, our analysis shows that the GPH results do not depend on the considered pressure level, further indicating the robustness of NZI and providing confidence on its physical underpinning. The similar correlations at various pressure levels also reveal that the GPH pattern has a barotropic structure”- I don’t think this section in its entirety is evident from any of the figures. Where is the analyses for multiple pressure levels shown?

We have followed your suggestion and we now present the analysis for different pressure levels in Fig S1 in the supplementary material, which shows the barotropic structure of the teleconnection.

Specific Comment 3

It would be interesting to see the lagged correlations between the NZI/ENSO indices and SWUS precipitation, with increasing lead times (perhaps starting a year prior) rather than averaging

climate indices over the preceding three to six months (i.e. just Jul-Sep and Sep-Nov) and then presenting the best results.

We have performed the suggested analysis and present the results in Fig. 4. Please see lines 143-161 for specific discussion of the results.

Specific Comment 4

Although the NZI yields relatively higher correlations with SWUS precipitation (Figure 1), the results from Figure S2 are not statistically robust enough to suggest the NZI can be utilized for useful predictions in a way that ENSO indices may not. Although a more compelling argument can be made for the period following 1970 (Figure 2b), specifically for fall; but I believe this will merit further research, including analysis.

In our study, we provide evidence for a new teleconnection and use ENSO as a “benchmark” teleconnection for comparison. We do not suggest excluding the use of ENSO indices as predictors for precipitation in SWUS and moreover, the NZI superiority relative to ENSO has appeared after the mid-1970s (e.g. see lines 29-31, 113-114, 127-141, and 254).

Specific Comment 5

The authors discuss the decadal variation of precipitation teleconnections and specifically point to a “periodicity of the order of 15 years,” but then elect to show a moving correlation analyses using only a 30-yr moving window. Could the authors comment on why they chose the 30-yr window? It would help to show the moving correlation across different windows (maybe ranging from 10-30 with 5 years’ increments).

Please note that correlation values based on 10 - 15 years cannot be reliably evaluated as to their statistical significance, due to the very limited sample size so the 30-year moving average window is the only viable option.

Specific Comment 6

Given the scarcity of data observation networks in the southern Hemisphere in comparison to the North, it will be of good value to add multiple reanalysis datasets for the most prominent analyses to serve as verification.

*We already used two different SST reanalysis datasets (see section **Methods and Data**). All results are almost identical based on both datasets, thus conclusions are robust. In the manuscript we now present results using both datasets to show robustness.*

Reviewer #2 (Remarks to the Author):

General Comment 1

The subject manuscript is interesting and well written, and identifying climate teleconnections beyond the established teleconnections is important and useful.

We thank the Reviewer for the encouraging comment.

General Comment 2

Although the paper has potential to be a useful contribution to the literature there are several issues that need to be addressed before the paper is suitable for publication. There are still several analyses that are needed to convincingly show the importance and reliability of the NZI.

We have performed extensive new analyses following your comments as detailed below and reflected in the revised manuscript. Point by point responses are provided below.

Specific Comment 1

The authors should consider referencing:

Newman, M., S.-I. Shin, and M. A. Alexander, 2011b: Natural variation in ENSO flavors. *Geophys. Res. Lett.*, 38, L14705, doi:10.1029/2011GL047658.

Johnson, N.C., 2013, How many ENSO flavors can we distinguish? *Journal of Climate*, 26:4816-4827.

Both studies are now cited. We have also added a discussion on the different ENSO flavors in the manuscript; see lines 122-126.

Specific Comment 2

Redmond and Koch (1991) performed one of the first studies of the predictive ability of ENSO indices. The authors should reference this highly cited paper.

Redmond, K. T., and R. W. Koch, 1991: Surface climate and streamflow variability in the western United States and their relationship to large-scale circulation indices. *Water Resour. Res.*, 27, 2381–2399.

We have added the suggested reference.

Specific Comment 3

The authors should provide an explanation for the selection of the 1950 through 2015 for analysis. Climate division data are available since 1895 and SST data go back to 1856. Why was the selected period chosen for this study?

Although reanalysis data are available since 1850s, SST data are not to be trusted in the period before 1950s due to the scarcity of the data observation networks, especially in the southern Hemisphere; see also [6-7]. Thus, we have decided to keep our analysis in 1950-2015 period. Please note also, that Referee 1 questioned even the accuracy of the pre-1970 southern hemisphere reanalysis SSTs which prompted us to report results from 2 reanalysis products.

Specific Comment 4

It would be helpful and interesting to compare the spectral frequencies of precipitation in the southwestern U.S. with the spectral frequencies of ENSO and NZI. What are the common spectral frequencies and do they change during the period analyzed?

We thank the Reviewer for his/her suggestion. As shown, in the following figure, due to the limited sample size, results are not very reliable.

To determine uncertainty we iteratively repeated the analysis after having disregarded each time a single value from the series. As expected, the highest uncertainty appears in higher frequencies.

As a general remark, for ENSO, important frequencies are those in the range of 0.15 to 0.35 yr^{-1} (periodicity 6.6 to 2.8 years), while for precipitation and NZI, lower frequencies are more important (higher time scales of periodicity; see also lines 108, 114, and Fig S2). Please note that due to the uncertainty of the results, we have decided not to include this figure in the manuscript.

Figure 2. (a) Spectral analysis of winter precipitation in SWUS (Nov-Mar) for 1950-2015. The results are calculated 66 times, each time disregarding a single value from the series. Solid line represents the mean of the ensemble, while shading the 95% intervals. (b)-(c) Same as (a) but for Niño 3.4 (Sep-Nov) and NZI (Sep-Nov), respectively. (d) Comparison of the means.

Specific Comment 5

Why was the region of the US used for the analysis restricted to the climate divisions with significant ENSO correlations? Also, why not look at precipitation correlations for the entire western U.S.? Does NZI result in a western U.S. precipitation dipole similar to ENSO?

Precipitation amounts over climate divisions which are located in the northern CA, NV and UT shows insignificant correlation with both NZI and ENSO indices. Indeed, precipitation in the northern side is known not to be related to Pacific SSTs, and to exhibit very different variability

than in the rest part of SWUS see Figs 1, 3, S10 and [1-5]. To account for this natural disassociation of the northern region, we define the regionally averaged SWUS precipitation as the area-weighted average precipitation amount over climate divisions for which ENSO (Niño 3.4) exhibits statistically significant correlations (significance level $\alpha = 0.05$; Figs 1a-b and 1h). Notice that defining the average precipitation based on the significance of NZI-correlations (vs. ENSO-correlations) does not affect the results (see Fig 1 herein) and moreover, it is not in the conservative side for comparing NZI and ENSO teleconnection strengths. Thus, we have clarified this issue by discussion in the manuscript; see lines 86-96.

We have also produced as suggested correlations of NZI and winter precipitation in the entire western US, and have added 2 new figures in the manuscript. Please see Fig. S9-10 and discussion in lines 194-199.

Specific Comment 6

A possible useful twist to this analysis would be to perform a couple of different analyses – (1) perform a principal components analysis of western U.S. precipitation and correlate the significant PC score time series with SSTs and see if the NZI index shows up in one of the correlation fields? (2) perform a spectral analysis of western U.S. precipitation and then correlate time series of significant spectral frequencies with SSTs and see if the NZI shows up.

We thank the Reviewer for his/her suggestion.

(1) We have performed the analysis. Results are provided in the supplementary material (see Figure S10). Concerning the PC2, SSTs in the ENSO region exhibit statistically insignificant correlation, while NZI exhibits a correlation value on the order of -0.6.

(2) As in the case of specific comment 4, the sample size is unfortunately too short to perform a reliable spectral analysis.

Specific Comment 7

Something that would really strengthen this paper is some analysis and explanation describing how the NZI affects atmospheric circulation and results in an effect on precipitation in the western U.S. How do changes in the southwestern Pacific move across the tropics and influence the mid-latitude atmospheric circulation of the Northern Hemisphere?

*We thank the Reviewer for his/her insightful comment and we have performed extensive new analysis to address it. In the new version of the manuscript we provide specific arguments as to the mechanism that NZI is associated with. This mechanism is now extensively described in section **The physical basis for the NZI teleconnection – The western Pacific pathway**, accompanied with several additional figures and results in the main text (Figs. 4-7) and in the supplementary material (Figs. S5-S10).*

Specific Comment 8

Do long-term trends in NZI influence the results? There is a long-term warming trend in the NZI that is not seen in NINO3.4 SSTs and this warming trend may be influencing the results. The authors should consider removing long-term trends in all data sets and re-computing the results.

To ensure that possible long-term warming does not affect our conclusions, we have reproduced the results of Fig. 1 c-f, after detrending SST and GPH time series at each grid point. As shown in the following figure, results are identical to those presented in Fig. 1 in the manuscript.

Figure 3. (a) Correlation map of SSTs (Jul-Sep) with winter precipitation in SWUS (Nov-Mar) for 1950-2015, after detrending time series. (b) Same as (a) but SSTs are averaged in Sep-Nov. (c)-(d) Same as (a)-(b), but using GPH data.

Reviewer #3 (Remarks to the Author):

General Comment 1

The authors present a not-well-known teleconnection and even less well known predictive measure for seasonal precipitation in the Southwest U.S. I believe the results will be useful to researchers and forecasters. Methods and analyses are appropriate, a few suggestions/questions appear below.

We thank the Reviewer for the encouraging comment.

General Comment 2

The fall NZI to winter SW precipitation relationship would be more meaningful if you were able to show how anomalous atmospheric circulation that actually produces above normal precipitation in the Southwest US evolves from a distant, un-connected SST and GPH anomaly over New Zealand.

*We thank the Reviewer for his/her insightful comment. In the new version of the manuscript we provide specific arguments as to the mechanism that NZI is associated with. This mechanism is now extensively described in section **The physical basis for the NZI teleconnection – The western Pacific pathway**, accompanied with several additional figures and results based on new analyses (see new Figs. 4-7 in main text, and Figs S5-S10 in supplementary material).*

Specific Comment 1

"72 SWUS, which is defined as the area-weighted average precipitation amount over climate divisions for which ENSO (Niño 3.4) exhibits statistically significant correlations (significance level $\alpha = 0.05$)" not standard definition of SW U.S. Explain why the ENSO connected region is adopted for purpose of testing NZI connection.

Precipitation amount over climate divisions which are located in the northern CA, NV and UT shows insignificant correlation with both NZI and ENSO indices. Indeed, precipitation in the northern side is known not to be related to Pacific SSTs, and to exhibit very different variability than in the rest part of SWUS see Figs 1, 3, S10 and [1-5]. To account for this natural disassociation of the northern region, we define the regionally averaged SWUS precipitation as the area-weighted average precipitation amount over climate divisions for which ENSO (Niño 3.4) exhibits statistically significant correlations (significance level $\alpha = 0.05$; Figs 1a-b and 1h). Notice that defining the average precipitation based on the significance of NZI-correlations does not affect the results (see Fig.1 herein), while it is not in the conservative side for comparing NZI and ENSO teleconnection strengths. We have added a discussion in the manuscript; see lines 86-96.

Specific Comment 2

"86 precipitation are shown in Figs 1c-d, for two lead times: Jul-Sept and Sept-Nov" lead times usually refer to the time lag between predictor and predictand. the Sept-Nov predictor is problematic in context of prediction because it overlaps with the Nov-Mar precipitation predictand.

We do not explicitly suggest that NZI averaged over Sep-Nov should be used as predictor for winter precipitation. Our investigation considers different periods to be complete. Indeed, for predicting purposes, one should consider NZI in a period that does not overlap with the period of interest of precipitation. We have changed our wording in the lines 98 and 449. Also, below we present results similar to those of Fig. 1, but when SST and GPH data refer to the period of Aug-Oct (no overlap with the period of Nov-March). It is obvious that NZI (Aug-Oct) is much correlated with precipitation (almost identical results with those in Fig 1), so conclusions are robust.

Figure 4. (a) Correlation map of SSTs (Aug-Oct) with winter precipitation in SWUS (Nov-Mar) for 1950-2015 (b) Same as (a), but using GPH (400mb) data.

Specific Comment 3

"90 the GPH results do not depend on the considered pressure level, further indicating the robustness of NZI and providing confidence on its physical underpinning." it is not unusual (actually expected) that multiple pressure levels vary coherently; multiple levels being involved does not indicate a robust association -- one being correlated essentially brings the others along.

We have deleted the sentence and we now present the analysis for different pressure levels in Fig S1 in the supplementary material, which shows the barotropic structure of NZI.

Specific Comment 4

"198 We quantified the effect of non-stationarity on the prediction skill as the relative increase of RMSE when the calibration and prediction periods did not overlap (e.g. prediction in 1983-2015, calibration in 1951-1983) relatively to the case that the calibration-prediction periods completely overlapped (e.g. prediction in 1983-2015, calibration in 1983-2015). " it is difficult to understand this; producing two separate correlations is a pretty meager cross validation.

We thank the Reviewer for his/her comment. We have changed our wording and added a statement to explain our results more effectively - see lines 237-240, and 246-248.

Specific Comment 5

Relevant references and findings that are not cited:

Cayan et al 1998 described. at decadal time scales, the association of SST and circulation in the western North and South Pacific, (including the New Zealand area) with anomalous precipitation in the SW U.S.

Dettinger et al. 1998 show both decadal and interannual trans-Pacific SST and sea level pressure patterns that accompany anomalous precipitation in western N. America, including a distinct SW US pattern.

Cayan, D.R., Dettinger, M.D., Diaz, H.F., and Graham, N., 1998, Decadal variability of precipitation over western North America: Journal of Climate, 11, 3148-3166.

Dettinger, M.D., Cayan, D.R., Diaz, H.F., and Meko, D., 1998, North-south precipitation patterns in western North America on interannual-to-decadal time scales: Journal of Climate, 11, 3095-3111.

Indeed, the suggested studies are very relevant and helpful. They are now both cited.

REFERENCES

1. Newman, M., *et al.* (2016) The Pacific Decadal Oscillation, Revisited, *J. Clim.*, 4399-4427.
2. Brown, D. P., and A. C. Comrie (2004) A winter precipitation “dipole” in the western United States associated with multidecadal ENSO variability, *Geophys. Res. Lett.*, **31**, L09203, doi:10.1029/2003GL018726.
3. McCabe, G. J. and M. D. Dettinger (1999), Decadal variations in the strength of ENSO teleconnections with precipitation in the western United States, *Int. J. Climatology*, **19**, 1399-1410.
4. Schonher, T. and S. E. Nicholson (1989) The relationship between rainfall and ENSO events, *J. Clim.*, **2**, 1258–1269.
5. Redmond, K.T., and R.W. Koch (1991) Surface climate and streamflow variability in the Western United States and their relationship to large-scale circulation indices, *Water Resources Research*, **27**(9) 2381-2399.
6. Seager, R., N. Harnik, Y. Kushnir, W. Robinson and J. Miller (2003) Mechanisms of hemispherically symmetric climate variability, *J. Clim.*, **16**(18), 2960-2978.
7. Seager, R., N. Harnik, W. A. Robinson, Y. Kushnir, M. Ting, H. Huang and J. Velez (2005) Mechanisms of ENSO-forcing of hemispherically symmetric precipitation variability, *Q. J. R. Meteorol. Soc.*, **131**, 1501-1527.

Reviewers' comments:

Reviewer #1 (Remarks to the Author):

The authors have made substantial effort in constructing a physical basis in support of their NZI connection to SW U.S. climate. Most of my earlier comments have been addressed. I have one remaining major comment and after this is addressed, I believe the paper can be considered for publication.

*The new section of "physical basis of the NZI teleconnection – The western Pacific pathway": The authors' main argument is that what's revealed in the NZI domain reflects a variation, or what appears to be expansion/shrinking, of the western Pacific warm pool. Fig. 4 depicts the cross-equator evolution of the SST anomalies that seem to originate from the SH region. In this framework, then the argument essentially falls back to teleconnection induced by the western North Pacific towards North America, a known process. Then, I'd like to point to a paper (doi: 10.1007/s00382-009-0722-5) in which a similar teleconnection has been found, one that resembles the trans-Pacific short-wave train of GPH authors show in Fig. 1. The present wave train resembles the non-ENSO wave train depicted in Fig. 4c of that paper (doi: 10.1007/s00382-009-0722-5). This is particularly relevant since the authors emphasize "decadal variation" in the teleconnections (e.g., line 104), like the quasi-decadal oscillation of that paper. Please reference.

Figure 6 indicates the jet stream shift, but how this affects the route or pattern of trans-North Pacific teleconnection is not well laid out. Jet stream shift alone can alter the stationary waves. This would seem to contradict (or fail to support) the tropical source of teleconnections. See e.g. Held et al. 2002: Northern Winter Stationary Waves: Theory and Modeling. *J. Climate*, 15, 2125–2144. Please consider either enhance the argument or delete this jet-related discussion.

Reviewer #2 (Remarks to the Author):

The authors have adequately addressed my previous concerns and questions regarding this paper. The addition of the explanation of the physical basis for the NZI teleconnection with SWUS precipitation is a good and necessary addition to the paper. However, the physical explanation is just a theory at this point. The physical explanation could be strongly supported if climate model experiments were used to confirm this theory.

Reviewer #3 (Remarks to the Author):

The Mamalakis et al. "Beyond ENSO.." paper is more clearly written and supported in its present version. I have a few comments (minor to a bit more important than minor):

line 128 cross validation was performed using data WITHIN a given 30 year window?
if so, this seems like you will obtain higher skill in the later period than is actually warranted (you already know the correlation is consistently high in the last 40 yrs or so of the record). a more honest way of assessing the skill is to use the entire record--as you note the non-stationarity may be natural variation, not necessarily an immutable climate change. and, accordingly, lines 135-141, reporting correlations for only the period since the 1970's without noting those for earlier period could lead reader to expect greater SWUS winter precipitation skill from NZI than is warranted--isn't it possible that in future the NZI linkage could revert to its former weaker status?

line 149 "DEPENDENCE of precipitation on the SST and GPH.." this is a statistical relationship, so more correct to term this an "ASSOCIATION of precipitation with SST and GPH"

line 182 and elsewhere "northeastern Pacific" might better be called "tropical and subtropical western North Pacific"

line 114 and legend to Figure S2 the 15 year periodicity is stated rather prominently but not really demonstrated (spectra). the record is relatively short to make a strong claim, which could be construed as characterizing a SWUS precipitation cycle.

Figure 4 to the eye, you are unfairly diminishing the positive correlations, especially in the ENSO core region of the late fall-winter maps. the color scale underplays positive correlations in the +0.4 to +0.5 category, compared to negative correlations -0.4 to -0.5. the faint salmon colored positive correlations hardly show up compared to the lighter blue negative correlations.

Figure 1 top you choose to plot correlations of N3.4 compared to those for NZI, but if you plotted SOI compared to NZI the ENSO relationship might look a bit stronger.

The ocean-atmosphere bridge mechanism--

this is an interesting idea. it seems useful to note that GCM experiments exploring seasonal atmospheric predictability (e.g. Hoerling, Kumar and others) have found much greater influence from SST in the tropical Pacific than from the extra-tropics. Your results propose an influence from SST outside the tropics it seems.

line 256 "and thus, they CAN be used towards increasing precipitation predictability."
i'd suggest saying "they MAY be used.....", since we don't know how long these stronger than ENSO linkages will hold up.

lines 59, 270 you refer to rainguage observations, which sounds like you use individual station precipitation records. but the divisional data are averages of precipitation from a set of stations whose membership has changed over time (depending on availability of date, etc). and "rainguage" might be interpreted as only including rain, but this is winter precipitation and important parts (higher elevations) of the SWUS divisions are dominated by snow during the winter.

line 177 and legend, Figure 5 " connecting 500 mb GPH anomalies in the area east TO Australia ..." do you mean "east OF Australia" ?

Reviewer #1 (Remarks to the Author):

The authors have made substantial effort in constructing a physical basis in support of their NZI connection to SW U.S. climate. Most of my earlier comments have been addressed. I have one remaining major comment and after this is addressed, I believe the paper can be considered for publication.

We thank the Reviewer for the encouraging statement.

*The new section of "physical basis of the NZI teleconnection – The western Pacific pathway": The authors' main argument is that what's revealed in the NZI domain reflects a variation, or what appears to be expansion/shrinking, of the western Pacific warm pool. Fig. 4 depicts the cross-equator evolution of the SST anomalies that seem to originate from the SH region. In this framework, then the argument essentially falls back to teleconnection induced by the western North Pacific towards North America, a known process. Then, I'd like to point to a paper (doi: 10.1007/s00382-009-0722-5) in which a similar teleconnection has been found, one that resembles the trans-Pacific short-wave train of GPH authors show in Fig. 1. The present wave train resembles the non-ENSO wave train depicted in Fig. 4c of that paper (doi: 10.1007/s00382-009-0722-5). This is particularly relevant since the authors emphasize "decadal variation" in the teleconnections (e.g., line 104), like the quasi-decadal oscillation of that paper. Please reference.

We thank the Reviewer for his/her comment. The study suggested by the Reviewer is indeed very relevant and helpful. It is now cited. The short-wave train mechanism was already cited, but not much highlighted. The teleconnection suggested by the Reviewer is now highlighted more in the manuscript (see lines 187-191).

Figure 6 indicates the jet stream shift, but how this affects the route or pattern of trans-North Pacific teleconnection is not well laid out. Jet stream shift alone can alter the stationary waves. This would seem to contradict (or fail to support) the tropical source of teleconnections. See e.g. Held et al. 2002: Northern Winter Stationary Waves: Theory and Modeling. J. Climate, 15, 2125–2144. Please consider either enhance the argument or delete this jet-related discussion.

We thank the Reviewer for his/her comment. Please note that results in Fig. 6 refer to a regional jet stream over a relatively narrow longitudinal zone centered at the west coast (220°E-260°E). It is not the zonally-averaged jet stream over the entire longitude. Zonal wind over this west coast area cannot be assumed to affect the stationary waves across the northern Pacific.

The teleconnection highlighted by the Reviewer (short-wave train) suggests how SST anomalies in the western Pacific alter the atmospheric pressure in the US west coast. Pressure anomalies affect the local wind and regional storm tracks and ultimately the winter precipitation. The regional jet stream we discussed in Figure 6 can be viewed as a result of this short-wave train teleconnection.

To address your concern and to make clear that we refer to the local upper wind field in Fig. 6, and also highlight the role of the atmospheric pressure anomalies in the physical process we have revised the wording appropriately (see lines 187-191, 209-211, 503-504).

Reviewer #2 (Remarks to the Author):

The authors have adequately addressed my previous concerns and questions regarding this paper. The addition of the explanation of the physical basis for the NZI teleconnection with SWUS precipitation is a good and necessary addition to the paper.

We thank the Reviewer for the encouraging statement.

However, the physical explanation is just a theory at this point. The physical explanation could be strongly supported if climate model experiments were used to confirm this theory.

We thank the Reviewer for his/her comment. Our argument of the strong relation between NZI and precipitation in SWUS is supported by rigorous analysis of observations and different reanalysis data sets which we present in Figs 1-4 and S1-4, S8-10. Concerning the physical explanation, all three steps of the mechanism that we propose are supported by results in Figs 4-6 and S5 -10. Moreover, based on our explanation, the increased teleconnection strength of NZI during the last 30-40 years is also supported in Fig. 7. Lastly, one should not disregard that each step in our proposed mechanism has already been reported partially in previous studies (see references in the main text). This provides an additional verification and physical justification of the NZI teleconnection. So overall, we believe that our study provides strong observational and physical evidence for one to claim that the suggested mechanism is more than “just a theory.”

We agree with the Reviewer that this new teleconnection and its mechanism should be further investigated and verified using idealized climate simulations and we have added a discussion in the conclusions (lines 258-261) to that effect. Such a modeling study however falls outside the scope of the current paper and is something we plan to pursue in the near future.

Reviewer #3 (Remarks to the Author):

The Mamalakis et al. "Beyond ENSO.." paper is more clearly written and supported in its present version. I have a few comments (minor to a bit more important than minor):

We thank the Reviewer for the encouraging statement.

line 128 cross validation was performed using data WITHIN a given 30 year window? if so, this seems like you will obtain higher skill in the later period than is actually warranted (you already know the correlation is consistently high in the last 40 yrs or so of the record). a more honest way of assessing the skill is to use the entire record--as you note the non-stationarity may be natural variation, not necessarily an immutable climate change, and accordingly, lines 135-141, reporting correlations for only the period since the 1970's without noting those for earlier period could lead reader to expect greater SWUS winter precipitation skill from NZI than is warranted— isn't it possible that in future the NZI linkage could revert to its former weaker status?

We thank the Reviewer for his/her comment. Yes, we use data within the 30yr window. Please note that Fig. S3 is presented as a supplementary material to the discussion of Fig. 2 which emphasizes the non-stationarity of the strength of the NZI and ENSO teleconnections within the 1950-2015 period. Fig S3 aims to strengthen the argument of the non-stationary nature of teleconnections by providing statistical measures other than correlation values, and evaluate the predictability of each index. Although the reader already knows that the correlation of NZI is high in the last decades, we believe that Fig S3 is a necessary addition to the discussion of Fig. 2.

Using all data to calibrate the linear models (as suggested) would mix the information of – what seems to be – two different statistical populations. Please note that the relation of NZI and precipitation is statistically insignificant in the first years of the record, and becomes statistically very strong and robust in the last decades, indicating that the teleconnection may have not always existed. Thus, although we present correlations for the entire period in Fig. 1, and the evolution of each relation in Figs 2 and S3, we are uncomfortable with the idea of performing the model calibration on the whole record since this would be statistically speaking “incorrect” given the known non-stationarity.

Concerning the last question and concern, please notice in lines 230-231, that we now explicitly state “... it is not certain whether the strength of the NZI signal will continue to increase or undergo periodic fluctuations in the future.”

line 149 "DEPENDENCE of precipitation on the SST and GPH.." this is a statistical relationship, so more correct to term this an "ASSOCIATION of precipitation with SST and GPH"

We thank the Reviewer. We have made the change both in the text and caption of Fig. 4.

line 182 and elsewhere "northeastern Pacific" might better be called "tropical and subtropical western North Pacific"

Thanks for the suggestion but we would prefer to retain the original simpler terminology.

line 114 and legend to Figure S2 the 15 year periodicity is stated rather prominently but not really demonstrated (spectra). the record is relatively short to make a strong claim, which could be construed as characterizing a SWUS precipitation cycle.

We thank the Reviewer. We have deleted the statement in the text and caption of Fig. S2.

Figure 4 to the eye, you are unfairly diminishing the positive correlations, especially in the ENSO core region of the late fall-winter maps. the color scale underplays positive correlations in the +0.4 to +0.5 category, compared to negative correlations -0.4 to -0.5. the faint salmon colored positive correlations hardly show up compared to the lighter blue negative correlations.

We thank the Reviewer. The colorscale used in Fig. 4 (and adopted for all correlation maps of the manuscript) is not intended to give unfair perception to the eye, especially since we present the real numbers of correlation coefficients throughout the manuscript. The correlations of ENSO are not higher than +0.5 in Figs 4e-f (we added a sentence in line 152-153), and this is why they hardly show up. Following your comment however, we tried using red color instead of salmon – no real difference is illustrated (see Fig. R1 below). In contrast, in Fig S5 for example, where high positive correlations exist, they are easily tracked. Note that in both figures, correlations within [-0.35,+0.35] are insignificant and are not shown.

Figure R1: Reproduction of Fig. 4e using different colorscale.

Figure 1 top you choose to plot correlations of N3.4 compared to those for NZI, but if you plotted SOI compared to NZI the ENSO relationship might look a bit stronger.

Please note that we present results and discussion on the relation of SOI and precipitation in two sections of the manuscript, and many figures/tables: Figs 2-3, S3 and Table S1. Specifically, in Fig. 1a,b, we chose to compare only SST-based indices and in particular only ENSO and NZI in order to demonstrate how and which SWUS climatic regions were chosen for our analysis.

The ocean-atmosphere bridge mechanism: this is an interesting idea. it seems useful to note that GCM experiments exploring seasonal atmospheric predictability (e.g. Hoerling, Kumar and others) have found much greater influence from SST in the tropical Pacific than from the extra-tropics. Your results propose an influence from SST outside the tropics it seems.

We thank the Reviewer. We have added a discussion in the conclusions (lines 258-261) and such analysis is planned to be pursued in the near future.

line 256 "and thus, they CAN be used towards increasing precipitation predictability." i'd suggest saying "they MAY be used.....", since we don't know how long these stronger than ENSO linkages will hold up.

We thank the Reviewer and we agree with the suggestion, which we have incorporated.

lines 59, 270 you refer to rainguage observations, which sounds like you use individual station precipitation records. but the divisional data are averages of precipitation from a set of stations whose membership has changed over time (depending on availability of date, etc). and "rainguage" might be interpreted as only including rain, but this is winter precipitation and important parts (higher elevations) of the SWUS divisions are dominated by snow during the winter.

We thank the Reviewer. This is a good point and we have reworded this sentence.

Line 177 and legend, Figure 5 " connecting 500 mb GPH anomalies in the area east TO Australia" do you mean "east OF Australia" ?

We thank the Reviewer. This was a typo and has been corrected.

Please consider the message of this rather long-ago paper by C. S. Ramage who emphasized the varying nature of large scale teleconnections: Ramage, C.S., 1983: Teleconnection and the Seige of Time, J. Climatol.3, pp. 223-31

We thank the Reviewer. This study is indeed helpful. It is now cited.

REVIEWERS' COMMENTS:

Reviewer #1 (Remarks to the Author):

I've read the replies and revision and do not have further comments.

Reviewer #3 (Remarks to the Author):

i am satisfied with the revisions and responses to comments i provided in my previous review. I have nothing to add and congratulate the authors in producing a new contribution to understanding and possibly predicting winter precipitation in the Southwest U.S.